# Peptide-Functionalized Nanoemulsions as a Promising Tool for Isolation and Ex Vivo Culture of Circulating Tumor Cells

**DOI:** 10.3390/bioengineering9080380

**Published:** 2022-08-10

**Authors:** Nuria Carmona-Ule, Noga Gal, Carmen Abuín Redondo, María De La Fuente Freire, Rafael López López, Ana Belén Dávila-Ibáñez

**Affiliations:** 1Roche-Chus Joint Unit, Translational Medical Oncology Group (Oncomet), Health Research Institute of Santiago de Compostela (IDIS), Hospital Gil Casares, 15706 Santiago de Compostela, Spain; 2Interdisciplinary Nanoscience Center (iNANO), Aarhus University, 8000 Aarhus, Denmark; 3Translational Medical Oncology Group (Oncomet), Health Research Institute of Santiago de Compostela (IDIS), University Clinical Hospital of Santiago de Compostela (SERGAS), 15706 Santiago de Compostela, Spain; 4Cancer Network Research (CIBERONC), 28029 Madrid, Spain; 5Nano-Oncology Unit, Health Research Institute of Santiago de Compostela (IDIS), University Clinical Hospital of Santiago de Compostela (SERGAS), 15706 Santiago de Compostela, Spain; 6DIVERSA Technologies S.L., 15782 Santiago de Compostela, Spain

**Keywords:** nanoemulsions, peptides, microfluidic, circulating tumor cells (CTCs), CTC isolation, on-chip cell culture

## Abstract

Circulating Tumor Cells (CTCs) are shed from primary tumors and travel through the blood, generating metastases. CTCs represents a useful tool to understand the biology of metastasis in cancer disease. However, there is a lack of standardized protocols to isolate and culture them. In our previous work, we presented oil-in-water nanoemulsions (NEs) composed of lipids and fatty acids, which showed a benefit in supporting CTC cultures from metastatic breast cancer patients. Here, we present Peptide-Functionalized Nanoemulsions (Pept-NEs), with the aim of using them as a tool for CTC isolation and culture in situ. Therefore, NEs from our previous work were surface-decorated with the peptides Pep10 and GE11, which act as ligands towards the specific cell membrane proteins EpCAM and EGFR, respectively. We selected the best surface to deposit a layer of these Pept-NEs through a Quartz Crystal Microbalance with Dissipation Monitoring (QCM-D) method. Next, we validated the specific recognition of Pept-NEs for their protein targets EpCAM and EGFR by QCM-D and fluorescence microscopy. Finally, a layer of Pept-NEs was deposited in a culture well-plate, and cells were cultured on for 9 days in order to confirm the feasibility of the Pept-NEs as a cell growth support. This work presents peptide-functionalized nanoemulsions as a basis for the development of devices for the isolation and culture of CTCs in situ due to their ability to specifically interact with membrane proteins expressed in CTCs, and because cells are capable of growing on top of them.

## 1. Introduction

Metastasis is one of the main causes of cancer death, and consists of a dynamic succession of events involving the dissemination of tumor cells [1]. These cells, called Circulating Tumor Cells (CTCs), are shed from the primary tumor and travel through the blood to colonize distant organs, generating a new metastatic niche [2]. Liquid biopsy refers to any technique for the detection and analysis of circulating tumor biomarkers in body fluids, most notably blood. Due to their characteristics, liquid biopsies allow physicians to monitor the blood instead of taking a tissue sample (i.e., conventional solid biopsy), providing patient information in real-time and in a less invasive manner. Various circulating biomarkers have been studied, such as exosomes, circulating cell-free DNA (ctDNA), functional messenger RNA (mRNA) [3], CTCs, and others [4]. However, among these biomarkers, CTCs are a powerful tool in research and clinical practice, mainly because they allow for analysis of the whole genetic material, thus providing in-depth knowledge about the disease and its evolution [5,6], and can be used in different functional downstream analyses [7].

Accordingly, molecular analysis of CTCs has become an outcome predictor in several types of cancer, including breast [8,9,10,11], prostate [12], colon [13,14], and bladder [15]. However, isolation of CTCs remains a challenge for clinical use, due to the rarity and heterogeneity of these cells. First, a very small number of CTCs is found in the blood, compared with blood cells (in a 7.5 mL tube of blood, it is unusual to identify more than 1–10 CTCs among billions of erythrocytes and millions of leukocytes). Second, there is a continuous evolution in their biomarkers, through epithelial–mesenchymal transition (EMT) or mesenchymal–epithelial transition (MET) [16,17].

Regarding CTC isolation, there are two main strategies based on: (1) their biological features (or label-dependent enrichment), and (2) their physical properties (or label-independent enrichment) [18,19]. When referring to their biological properties, the most relevant technology is CellSearch^®^ (Menarini Silicon Biosystems, Inc., Bryn Athyn, PA, USA). This system is the only device to have been approved by the US Food and Drug Administration (FDA) for clinical application in prostate, colorectal, and breast cancers (Parsortix^®^, ANGLE, United Kingdom and Toronto, Canada, has also recently been approved), and consists of epithelial cell adhesion molecule (EpCAM) antibodies linked to magnetic beads [20]. However, its main disadvantage is that cells expressing low levels of EpCAM may not be captured [21]. Related to physical properties, there are several strategies, including size- (Parsortix^®^) or density-based approaches (OncoQuick^®^, Greiner BIO-ONE, Madrid, Spain). Despite the advances in the new technologies marketed in recent years, there is still no existing technology based on either biological or physical properties that can be applied as a standard method for the isolation and culture of CTCs. To address this issue, researchers are moving toward the use of combinations of both characteristics (i.e., biological and physical) in the same device through the use of microfluidic strategies such as the Target Selector™ (Biocept, Inc., San Diego, CA, USA) platform, IsoFlux (Fluxion Bioscience, Inc., Alameda, CA, USA), or HBCTC-Chip [22]. Microfluidics appears is a very interesting approach, due to the associated inner features that allow for easy multi-function integration, continuous sample processing, a reduction in sample consumption, and changeable chip design, leading to high-speed, high specificity, high-throughput, and easy operation characteristics [23]. In fact, due to the advancement of microfabrication and nanomaterials, wide ranges of methodologies have emerged for the enrichment of CTCs on microfluidic platforms [24]. In order to improve the drawbacks related to the limitations of CTC isolation technologies, we developed a bottom-up methodology to formulate Peptide-Functionalized Nanoemulsions (Pept-NEs) targeting cancer cells. Our NEs have been previously shown to favor the proliferation of breast cancer cells and CTCs isolated from patients in culture [25,26]. Hence, the aim of this work was to develop a formulation which is able to isolate CTCs and culture them in a single step through the use of a tailored microfluidic device and exploiting the favorable effect observed on CTC cultures when supported with NEs.

As a proof of concept and to validate our isolation technology, we use immune recognition to specifically isolate cells expressing two specific antigens—EpCAM [27] and EGFR [28]—which have been thoroughly studied as biomarkers for the targeting of breast cancer cells [29,30].

At present, most CTC isolation technologies are based on nanomaterial surfaces decorated with antibodies; however, in this paper, we use peptides as a novel approach. Compared with antibodies, peptides are stable, small, and easy to synthesize in large amounts [31]. Consequently, we selected two peptides to functionalize our NEs, GE11 and Pep10, which recognize EGFR and EpCAM, respectively.

The peptide Pep10 was identified by Wang through the use of cell-based selection in 2014 [32], and has proved to effectively isolate EpCAM+ cells by the functionalization of magnetic nanoparticles for spiked breast, prostate, and liver cancer cells from human blood. Additionally, they proved that Pep10 is a good candidate for CTC isolation and as a surface modification ligand—as it showed comparable binding affinity to EpCAM antibody—for EpCAM recognition (Pep10 KD is 1.98 × 10^−9^ mol/L and the EpCAM antibody KD is 2.69 × 10^−10^ mol/L) [32].

GE11 peptide affinity for EGFR targeting has been identified by screening of phage display libraries [33]. Additionally, its efficiency to target EGFR has been shown using the GE11 peptide-modified polymersomal doxorubicin, which has emerged as an advanced alternative to the current Lipo-Dox treatment in EGFR over-expressing ovarian cancers [34]. Moreover, this peptide has been used to capture liver CTCs using lipid bilayers decorated with EGFR, showing higher CTC capture for hepatic carcinoma cells than Cell-Search^®^ [35].

Therefore, the main hypothesis of this work is that, by using the proposed Pept-NEs as the basis of a CTCs isolation device, we may address the most challenging elements in the field of liquid biopsy: (1) isolation of CTCs, and (2) ex vivo culture.

Additionally, this approach has the potential to provide a versatile platform where, in the future, different peptides could be added to functionalize the NEs. The workflow for this study is represented in Figure 1.

## 2. Materials and Methods

### 2.1. Materials

#### 2.1.1. Reagents and Solvents

Lipoid^®^ S100 PC (18:0/18:1) from soybean (94%) was a gift from Lipoid GmbH (Ludwigshafen, Germany). Cholesterol Polyethylene glycol N-Hydroxysuccinimide (CH PEG NHS; MW: 2 kDa) was purchased from Nanocs Inc. (New York, NY, USA). OA, CH, TopFluor^®^ PC, Dimethyl Sulfoxide (DMSO), Polylysine (PLL, MW 30–70 kDa), Cell Counting Kit-8 (CKK-8), trypsin with EDTA solution, acetic acid solution, FBS, BSA, and penicillin-streptomycin were purchased from Merck KGaA (Sigma-Aldrich^®^, Darmstadt, Germany). Ethanol (absolute) was purchased from Scharlab, S.L (Barcelona, Spain). Amicon^®^ Ultra-4 100 K centrifugal filters were provided by Merk (Millipore^®^ Darmstadt, Germany). MQ-water was purified by Millipore^®^ Direct-Q^®^ 3 with a UV system. dPBS and High-glucose DMEM were purchased from Biowest (Nuaillé, France). Tissue culture dishes (100 mm) were provided by VWR International, LLC (Avantor^®^, Barcelona, Spain). Black 96-well plates and ultra-low-attachment 24-well plates were purchased from Corning Inc. (Corning, NY, USA). Paraformaldehyde solution (PFA) 4% in PBS and NucBlueTM Hoechst 33,342 dye were purchased from Thermo Fisher Scientific, (Waltham, MA, USA). Human EGFR protein (His Tag-Biotinylated; 630 aa, MW: 69.8 kDa) and Human EpCAM protein (His Tag-Biotinylated; 253 aa, MW: 29 kDa) were provided by Sino Biological Inc. (Eschborn, Germany). All peptides were purchased from GenScript, (TWIN HELIX SRL, Rome, Italy): GE11 (12aa: YHWYGYTPQNVI), fluorescent GE11 (13aa; sequence: YHWYGYTPQNVI {K (TMR)}), Pep10 (23aa; sequence: VRRDAPRFSMQGLDACGGNNCNN), and fluorescent Pep10 (24aa; sequence VRRDAPRFSMQGLDACGGNNCNN {K (TMR)}).

#### 2.1.2. Cell Cultures

All the cell lines used in this paper were purchased from Merck (Sigma-Aldrich^®^, Darmstadt, Germany). The human breast cancer MCF-7 and MDA-MB-231 cell lines were cultured in high-glucose DMEM. The cell culture medium was supplemented with 10% fetal bovine serum and 1% penicillin-streptomycin. Cells were cultured at 37 °C in a humidified atmosphere containing 5% CO_2_. At 85% confluence, cells were harvested using 0.05% Trypsin-EDTA (5 min, 37 °C).

### 2.2. Methods

#### 2.2.1. Formulation of PEGylated Nanoemulsions (PEG-NEs)

PEG-NEs with cross-linking reagent (NHS ester) on the surface were formulated at a 1 mL scale by a low-energy self-emulsification oil in water (O/W) method. Briefly, stock solution of lipids PC (10 mg/mL), CH (10 mg/mL), CH linked to a polyethylene glycol chain-terminated N-Hydroxysuccinimide (CH PEG NHS; 10 mg/mL), and OA (100 mg/mL) were prepared in ethanol. The concentration of the stock solution of CH PEG NHS used in the formulation was calculated following protocol recommendations for NHS ester reactions. In case of fluorescently labelled formulations, a stock solution of TopFluor^®^ PC (1 mg/mL) was additionally used. Then, lipids were mixed to obtain an organic phase at final molar percentages of 3.45% PC, 3.38% CH, 0.65% CH PEG NHS, 92.48% OA, and 0.05% TopFluor^®^ PC. The organic phase (100 μL) was quickly injected into deionized water (900 μL) under magnetic stirring at room temperature. After 15 min, stirring was stopped and the PEG-NEs with the NHS ester surface ending were obtained. The PEG-NEs formulation was concentrated and purified using one Amicon^®^ Ultra-4 100 K filter at 3234× *g* for 29 min at room temperature (Figure 2).

#### 2.2.2. Conjugation of GE11 and Pep10 Peptides onto Surface of PEG-NEs

To endow PEG-NEs with active targeting capability to CTCs, two peptides (GE11 and Pep10) were conjugated to obtain the Pept-NEs: GE11-NEs and Pep10-NEs, respectively. First, peptide stocks were prepared at 0.5 mg/mL. For Pep10, the solvent used was MQ-water, while for GE11 a solution of ratio 1:1 of MQ-water and 10% acetic acid was used. In brief, previously synthesized PEG-NEs were incubated with Pep10 (0.0195 mg/mL) or GE11 (0.0163 mg/mL) in 1 mL of dPBS (pH 7.4) for 30 min at room temperature under magnetic stirring (Figure 2). The resulting GE11-NEs and Pep10-NEs were isolated from the non-conjugated peptides using Amicon^®^ Ultra-4 100 K filters by centrifugation at 3234× *g* for 20 min (room temperature conditions). The filters allowed us to retain components bigger than 100,000 Da, such as PEG-NEs, while components with lower sizes, such as the peptides (GE11, 1377 Da; Pep10, 2496 Da) and free lipids passed though the filter. Then, both formulations were washed with MQ-water twice, using the same Amicon^®^ Ultra-4 100 K filters, followed by centrifugation at 3234× *g* for 20 min at room temperature.

#### 2.2.3. Characterization of Pept-NEs

##### Dynamic Light Scattering (DLS) and Nanoparticle Tracking Analysis (NTA)

Mean average size (z-average) and PdI were determined by DLS at room temperature using a Zetasizer^®^ (Nano ZS90, Malvern Panalytical Ltd., Grovewood, UK). The ζ -Potential was determined using electrophoretic cells (Malvern Panalytical Ltd., Madrid, Spain). The concentrations of GE11-NEs and Pep10-NEs were determined by NTA using a NanoSight NS300 instrument (Malvern Panalytical Ltd., Madrid, Spain). All measurements were conducted using a 1:100 *v*/*v* dilution ratio in MQ-water. The stability of Pept-NEs stored at 4 °C was measured (size, PdI, and ζ-Potential data were acquired once a month for a period of 3 months).

##### Transmission Electron Microscopy (TEM) Analysis

Pept-NEs were stained to prepare the TEM grids. For this, 25 µL of GE11-NEs and Pep10-NEs were mixed separately with 25 µL of 2% sodium phosphotungstate. Once the Pept-NEs were negatively stained, a drop of each of the samples was plunged on copper grid films for 2 min. After washing with MQ-water and drying overnight in vacuum conditions, the samples were observed using the TEM (JEOL 2010 TEM) of the Technological Scientific Support Center (CACTUS) at the University of Santiago de Compostela.

#### 2.2.4. Efficiency of Peptide Functionalization on Pept-NEs Formulations

Fluorescent Pet-NEs were used to calculate the association efficiency of the peptides onto PEG-NEs surfaces. The fluorescently labelled peptides GE11 (TMR) and Pep10 (TMR) were used during the formulation process of Pept-NEs to obtain Pept(TMR)-NEs. These florescent peptides were additionally used to prepare the standard curves of the peptides GE11 and Pep10. First, the fluorescence intensities of GE11(TMR) and Pep10(TMR) at increasing known concentrations were read on a FLUOstar OPTIMA (BMG LABTECH, Ortenberg, Germany) multi-mode plate reader in the presence of a constant concentration of PEG-NEs. Then, the standard curve of each peptide was represented (fluorescence intensity vs. concentration) and the equation was extracted. We decided to add the aforementioned fixed concentration of PEG-NEs to perform the calibration in order to extract the scattering noise arising from the nanoemulsions. Next, once the standard curves of the peptides were obtained, the fluorescence intensities of the Pept(TMR)-NEs samples—that is, the GE11(TMR)-NEs and Pep10(TMR)-NEs—at the same NP concentration employed for calibration curve creation were read using the FLUOstar equipment. Finally, the unknown concentration of each of the peptides in the Pept(TMR)-NEs samples were determined by applying the equation obtained from the standard curves. The measurement conditions were excitation wavelength, 544 nm and emission wavelength, 590 nm.

#### 2.2.5. Quartz Crystal Microbalance with Dissipation Monitoring (QCM-D) Experiments

QCM-D experiments (Q-Sense E4, Biolin Scientific, Gothenburg, Sweden) were performed to both analyze the adsorption behavior of the NEs onto different surfaces (immobilization) and to study the specific binding of the Pept-NEs to their receptors. Before starting any experiment, QCM-D crystals were cleaned by immersion in a 2% sodium dodecyl sulphate solution overnight, then rinsed with MQ-water. Then, crystals were blow-dried with N_2_ followed by exposure to UV for 20 min. For all experiments, frequency changes (ΔF) and dissipation changes (ΔD) were monitored at 24 ± 0.02 °C.

First, to determine the optimal surface and buffer required for the immobilization of PEG-NEs onto surfaces, three different surfaces were tested: (1) gold-coated crystals (QSX301, Q-sense), (2) silica-coated crystals (QSX300, Q-sense), and (3) PLL pre-coated silica crystals (QSX300, Q-sense). Furthermore, three different buffers were used: (A) MQ-water, (B) dPBS, and (C) HEPES2 (10 mM HEPES, 150 nM NaCl; pH 7.4). In the particular condition of the PLL pre-coated silica crystal, when a stable baseline in buffer solution was achieved, the polymer solution (1 mg/mL of PLL in the corresponding buffer) was introduced into the measurement chamber and left to adsorb onto the silica crystal. In this case, after the surface was saturated, the chamber was rinsed with buffer solutions to remove the excess polymer. Once pre-coated with PLL, this surface and the other two (Au and silica crystals) were exposed to negatively charged PEG-NEs and incubated until the surfaces were saturated. The PEG-NEs were introduced in the chambers using a 1:2 *v*/*v* dilution ratio in the corresponding running buffer. Then, the buffer solution for each condition was introduced for washing. After that, PEG-NEs layer stabilities were analyzed under different conditions.

Second, to study the specific binding ability of the Pept-NEs against their targets, only PLL pre-coated silica crystals and MQ-water (as running buffer) were used. First, the PLL layer was deposited onto a previously exposed silica surface (1 mg/mL PLL, MQ-water), then solutions of GE11-NEs and Pep10-NEs were exposed separately to the PLL layer and left to incubate until the surfaces were saturated (both formulations were introduced using a 1:2 *v*/*v* dilution ratio in MQ-water). The Pept-NEs solutions introduced into the chambers were replaced with MQ-water to wash away the Pept-NEs that had not linked to the PLL surfaces. Then, the chambers were loaded with EpCAM or EGFR protein solutions at a concentration of 0.8 µg/mL (MQ-water). Finally, MQ-water was used to rinse the unlinked peptides. All experiments were carried out in duplicate under a continuous flow (150 µL/min). The third overtone was used to present normalized frequencies. The increase in mass on the surface (adsorption effect) was reflected by a negative ΔF (i.e., bars point upward), while a desorption effect or any phenomenon leading to mass loss resulted in a positive ΔF (i.e., bars point downward).

#### 2.2.6. Fluorescence Microscopy

First, the specific expressions of EpCAM by MCF-7 and EGFR by MDA-MB-231 were validated. Cells were washed with dPBS (Lonza, Basel, Switzerland) and fixed with cold 4% paraformaldehyde (PFA; ThermoFisher Scientific, Waltham, United States) for 15 min. Then, cells were washed with dPBS (Lonza). Afterwards, cell solutions were immunostained using EpCAM antibody (dilution 1:50, APC, Clon HEA-125, Miltenyi, Bergisch Gladbach, Germany) and EGFR antibody (dilution 1:50, Alexa 488, Clone LA1, Merck, Darmstadt, Germany). NucBlueTM (Hoechst 33,342 dye, ThermoFisher, Waltham, MA, USA) was used to counterstain nuclei. InsidePerm Buffer (Inside Stain Kit, Miltenyi Biotec, Bergisch Gladbach, Germany) was used as an antibody diluent. Stained solutions were examined by fluorescence microscopy (Leica DMi8 automated Microscope, Leica Microsystems S.L.U., Barcelona, Spain), for which a 63× oil immersion objective was used. Cell targeting was also analyzed by fluorescence microscopy. Therefore, fluorescently labelled Pept-NEs and bare PEG-NEs were formulated using TopFluor^®^ PC, and the specific binding of Pep-NEs for their targeted cells was assessed. MCF-7 and MDA-MB-231 were seeded at a final concentration of 30,000 cells/mL at 0.2 mL per well (ibidi 8-well plates) and left to attach overnight. Then, cells were fixed using 4% cold PFA in PBS (15 min, room temperature) and washed three times. Once fixed, they were left to interact with PEG-NEs, GE11-Nes, or Pep10-NEs for 30 min at room temperature. In the negative control, PBS was used instead. Next, samples were washed three times with PBS, and NucBlueTM (Hoechst 33,342 dye) was added for staining of nuclei. A Leica DMi8 automated Microscope (Leica Microsystems) was used, equipped with corresponding filters to take the images. Again, a 63× oil immersion objective was used.

#### 2.2.7. Cell Viability Analysis

A K-8 kit was used to analyze the viability of the cells incubated with the immobilized formulations (PEG-NEs, GE11-Nes, and Pep10-NEs); PLL and the wells without adding any extra layer (Ø) were used as controls. First, 1 mg/mL of PLL solution in MQ-water was added to each well (only in the conditions PLL, PEG-NEs, GE11-Nes, and Pep10-NEs). This PLL solution was incubated for 5 min and washed twice using MQ-water. Second, stock formulations (PEG-NEs, GE11-Nes, and Pep10-NEs) were added into the wells with a 1:2 *v*/*v* dilution ratio in MQ-water (50 µL/well). In the same way, conditions Ø and PLL were only treated with 50 µL/well of water. After 1 h of incubation, wells were washed twice with MQ-water, then washed with DMEM-HG. Then, MCF-7 and MDA-MB-231 in DMEM-HG were seeded in these pre-treated plates (1000 cells/well in 100 µL). The measurement times were 0, 1, 3, 6, and 9 days (denoted as t0, t1, t3, t6, and t9, respectively). Media were refreshed every 3 days. For all the time measurements, 10 µL of CCK-8 was added to each well and cells were incubated for 2 h at 37 °C and 5% CO_2_. Finally, 100 µL of the solution from each well was transferred to a new plate and the absorbance was measured using a multi-mode plate reader (Infinite M1000, Tecan Group Ltd., Männedorf, Switzerland) at 450 nm. Three independent repeats were performed for all conditions.

#### 2.2.8. Statistical Analysis

Statistical analysis was performed using the GraphPad Prism 6.01 software (GraphPad Software Inc., La Jolla, CA, USA). Student’s t-test was used for the comparison between two group analyses. In cases of *p*-values less than 0.05, the result was considered statistically significant.

## 3. Results

The originally formulated NEs published in our previous work were modified by the addition of CH-PEG with an N-Hydroxy Succinimide ester (NHS) ending group into the standard formulation (PEG-NEs). The modification of NEs allowed for the binding of GE11 and Pep10 peptides onto the PEG-NEs surface, thus obtaining GE11-Nes and Pep10-NEs, respectively. The NHS ending group on the NEs surface facilitated a covalent reaction between the ester (NHS) and the amino tail groups from the peptides, in basic conditions (pH 7.4), after a short incubation time (30 min). The formulations were characterized, in terms of hydrodynamic sizes for PEG-NEs, GE11-NEs, and Pep10-NEs (Figure 3B). NTA was performed for Pept-NEs. All formulations were less than 200 nm with a PdI lower than 0.2, suggesting good monodispersity for the three formulations. Furthermore, their ζ-Potentials were measured, showing negative surface charges for all the formulations.

In addition, TEM imagery demonstrated that the two Pept-NEs had spherical shapes (Figure 3D). At the same time, the Pept-NEs concentrations (NPs/mL) were determined by NTA, proving the reproducibility of the synthesis process to obtain Pept-NEs, as the concentrations of both of the Pept-NEs remained similar. The final concentrations of the peptides anchored to the NEs were calculated using the standard curves of fluorescently labelled peptides (Figure 3E and Appendix A). Furthermore, we evaluated that the formulations remained stable for at least three months (Appendix A).

### 3.1. PEG-NEs Immobilization Analysis onto Different Surfaces by QCM-D

In order to determine the optimal conditions for immobilization of the NEs on chip surfaces, we first analyzed the adsorption behaviors of the intermediary PEG-NEs (before peptide functionalization) using a label-free methodology: QCM-D. For this experiment, we used the intermediate NEs (PEG-NEs) as our final goal was to obtain versatile NEs which can be modified with any peptide to target various proteins of interest. The tested crystals were gold crystals, silica crystals, and pre-coated PLL silica crystals. To optimize and quantify the concentration of bare PEG-NEs incorporated on the QCM crystals, the frequency changes of these different QCM crystals were analyzed upon the addition of PEG-NEs using three buffers: MQ-water, dPBS, and HEPES2. In the case of PLL pre-coated silica crystals, a PLL solution (1 mg/mL, in each of the three buffers) was introduced into the measurement chamber until a stable baseline was achieved. Then, the PLL solution was left to adsorb onto the crystal to form a polymer precursor layer. After the surface was saturated with PLL, the chamber was rinsed with the specific buffer solution, to remove the excess polymer. Once the surface was pre-coated with PLL, all surfaces employed (gold crystals, silica coated crystals, and PLL pre-coated silica crystals) were exposed to the negatively charged PEG-NEs and left to incubate until surface saturation. Figure 4 shows the results of PEG-NES immobilization on the three surfaces, assessed for the different solvents tested. The effective surface coating (adsorption of polymer/PEG-NEs) is expressed as an increase of mass on the crystal surface showing a negative ∆F (the bar points upwards). By contrast, PEG-NEs desorption after washing led to a mass loss, resulting in a positive ∆F (the bar points downwards). Therefore, as can be observed from Figure 4, the condition employing pre-coated PLL silica surface and MQ-water as solvent produced the most stable monolayer of PEG-NEs, as no desorption was observed after washing them, when compared with the two other solvents (dPBS and HEPES), where the coverage was only half. In the case of silica crystals, for any of the solvents tested the best of the conditions was not as good as that for the pre-coated PLL silica crystals. Moreover, even though an Au surface with HEPES as the solvent seems to be a good option for developing a surface to deposit monolayer PEG-NEs, as no desorption was observed this condition was rejected due to the large error bars. Therefore, PLL pre-coated silica crystals and MQ-water as buffer were chosen as the conditions for perform the following experiments to assess protein targeting by Pept-NEs.

### 3.2. Assessment of Pept-NEs Binding to Their Specific Proteins under Continuous Flow, Analyzed by QCM-D

To validate the specific recognition of the considered Pept-NEs for their targeted proteins, a QCM-D technique was used. As previously mentioned, the GE11 recognition peptide targets EGFR and Pep10 targets EpCAM (Figure 5A). Based on the previous results obtained, we chose PLL-coated silica crystals and MQ-water for the Pept-NEs immobilization experiment. Once Pept-NEs were immobilized on PLL-coated silica crystals, protein solutions in MQ-water (EGFR or EpCAM) were introduced separately into the chambers. We studied the binding affinity under continuous flow conditions (150 µL/min). As shown in Figure 5B, Pep10-NEs showed the highest binding ability to EpCAM recombinant protein, while GE11-NEs presented very low binding capacity under the same conditions. This result proves that Pep10-NEs can selectively bind to EpCAM. Surprisingly, related to EGFR, from Figure 5B, it is possible to observe desorption or other phenomena leading to mass loss in both cases (Pep10-NEs and GE11-NEs) after washing. The desorption observed might be related to the higher molecular weight of the EGFR protein.

### 3.3. Specific Binding Analysis between Pept-NEs and Cells by Fluorescence Microscopy

To determine the specific recognition of Pept-NEs for breast cancer cells expressing different biomarkers, fluorescently labelled Pept-NEs (using TopFluor^®^ PC) were formulated to follow their distribution by fluorescence microscopy. Additionally, fluorescently labelled PEG-NEs were formulated as a negative control. The cell lines used to perform this experiment were MCF-7 and MDA-MB-231, as we have observed in previous works [36,37] and in our group that MDA-MB-231 expresses EGFR and MCF-7 expresses EpCAM. Additionally, to confirm this protein expression, we performed immunohistochemical analysis on both cell lines (Figure 6A). The data demonstrated that MCF-7 expressed EpCAM but not EGFR, whereas MDA-MB-231 expressed EGFR but not EpCAM. Therefore, the MCF-7 and MDA-MB-231 cell lines were selected to confirm the specific binding ability between the Pept-NEs and the specific cell populations. As shown in Figure 6B, the Pep10-NEs presented selective interaction with the EpCAM^+^ MCF-7 cells but not MDA-MB-231. Likewise, GE11-NEs demonstrated specific biding to EGFR^+^ from MDA-MB-231 but not MCF-7, indicative of the ability of these Pept-NEs to target their specific receptors. Therefore, the results show that Pept-NEs preferentially bind to their target cell.

### 3.4. Cell Viability Assessment of Immobilized Pept-NEs

In order to prove that cells can grow on the Pept-NEs once isolated, a viability assay was performed. The experimental set-up is represented in Figure 7A; the concentration employed was the same as in QCM-D experiments. To mimic the situation of CTCs in a blood sample from patients, the smallest number of cells that the CCK-8 can detect were seeded (1000 cells/well). As is shown in Figure 7B, non-toxicity was observed in all conditions after 9 days of culturing.

## 4. Discussion

Numerous attempts have been made to isolate and culture CTCs with the aim of analyzing them. The main problem hindering successful culture lies in the relevant isolation technologies that compromise the viability of the CTCs. Therefore, based on our previous works, in which we showed the effectiveness of NEs [25,26] in enhancing CTC cultivability, we proposed their use as the basis for an isolation and culturing chip.

We successfully functionalized NEs with two different peptides, Pep10-NEs and GE11. To perform functionalization, our previously published NEs [26] were modified with CH-PEG having an N-Hydroxy Succinimide ester (NHS) ending group. To perform the reaction between the NEs and peptides, we conducted cross-linking between the ester from the surface of NEs and the amino tail groups of the peptides. We selected NHS as the cross-linker, as it is one of the most commonly used cross-linkers to obtain a strong covalent chemical attachment between peptides and a primary amine [38,39]. This reaction has been previously employed for the generation of phospholipid–polyethylene glycol-derived functional conjugates for molecular imaging and targeted therapy [40]. It is important to note that in order to achieve success in the functionalization process, we needed to control two main factors: (a) pH, in order to ensure conjugation between the carboxylic end group of NE and the amino group of the peptides, and (b) purification, to ensure that all peptides become part of the formulation, rather than remaining free in solution.

Characterization of the Pept-NEs confirmed their good monodispersity. Furthermore, through the analysis of changes in ζ-potential, we proved the effectiveness of the peptide functionalization. We observed that GE11-NEs showed a more positive ζ-potential value than the intermediary PEG-NEs. This is in agreement with previous works, which have shown how GE11 increases the surface charge of formulations due to the positive overall charge of the GE11 peptide [41]. Oppositely, for Pep10-NEs, as Pep10 possesses a negative overall charge, the Pept-NEs showed a more negative ζ-Potential than PEG-NEs.

Next, we used a label-free technology—Quartz Crystal Microbalance with Dissipation (QCM-D) analysis—to study the immobilization of PEG-NEs onto surfaces and the specific binding of Pept-NEs by their protein targets. This assay was performed with the intermediate NEs (PEG-NEs), as our final goal was to obtain versatile NEs which can be modified with any peptide in order to target various proteins of interest. Based on the obtained results, we established that a pre-coated PLL silica surface is the most best for attaching the PEG-NEs. Our data are in agreement with previous works that have reported that the functionalization of a mica surface with positively charged PLL promoted the adsorption of DNA and proteins [42]. Moreover, this PLL surface ensures biocompatibility for future chips, as PLL is a well-studied polymer for biomedical applications [43].

We also proved that Pep10-NEs and Pep-GE11 presented binding ability for their recombinant proteins—EpCAM and EGFR, respectively—using QCM-D. It is important to mention that, in the case of EGFR, we observed a desorption process when bound to both Pept-NEs after the washing process. We associate this fact with the higher molecular weight of this protein, as EGFR has a MW of 69.8 kDa, making it much bigger than EpCAM (with a MW of 29 kDa). Additionally, it is well-known that protein conformation influences inding affinity by affecting the free binding sites [44]. Similarly, it has been shown that the anion specific conformation can be used to control protein adsorption/desorption on the surface of PSBMA brushes [45]. Based on this, the conformations of EpCAM an EGFR can differ, depending on different biological conditions [46,47]. These differences in conformations could influence the binding strength between EGFR and Pept-NEs which, in addition to its higher molecular weight and the experimental flow conditions chosen, could explain the desorption process observed for EGFR. In any case, GE11-NEs presented the higher desorption effect, suggesting a strong interaction with their targeted protein (EGFR).

Furthermore, we validated the capability of Pept-NEs to recognize their specific proteins using two different cell lines—MCF-7 and MDA-MB-231—which express EGFR an EpCAM, respectively. We observed the selective interaction of Pep10-NEs to EpCAM+ MCF-7 cells, but not MDA-MB-231. Likewise, GE11-NEs demonstrated specific biding to EGFR+ from MDA-MB-231, but not MCF-7. This experiment corroborated the tendency of the Pept-NEs to target their specific receptors. We are aware that the number of CTCs obtained from cancer patient samples will be much lower than the number of cells used in these experiments; nevertheless, we have proven the specificity of our Pept-NEs for two well-studied CTC biomarkers: EpCAM, the cell adhesion molecule used by CellSearch platform, and EGFR, which may be used as an alternative to EpCAM, as its expression has been confirmed in CTCs [48]. We believe that these results lay the foundation for the development of a future device using Pept-NEs to isolate CTCs.

Finally, since the final goal of the system is the culture of isolated cells in an isolation device, we performed viability assays, which showed no cytotoxicity for any of the conditions assessed. Based on our previous results, where we proved a proliferative effect with the addition of NEs to cultures [26], here, we also expected to obtain a proliferative (increment in viability) effect in the case of the cells cultured on top of Pept-NEs and PEG-NEs. However, the data did not show any increment in viability. This could be explained by the fact that the concentration used for NE deposition in the plates was not the same as that used for the experiments performed in our previous work; therefore, the amount of lipids and fatty acids endocytosed by cells was not sufficient to achieve cell metabolism activation. Thus, more work should be done to overcome these issues. Additionally, this experiment must be repeated using a lower initial concentration of cells to mimic a realistic CTCs situation, even though this is a first approximation to validate the cultivability of cells in a plate covered by Pept-NEs.

This work aimed to improve the most common challenge after the isolation of CTCs from patient’s blood, namely, their release from the capturing substrate at the end of the process. We propose an on-chip culture to allow for the proliferation of CTCs, reducing the risk of cell loss to increase the success rate of CTCs culturing. On one hand, the presence of Pept-NEs on chip surfaces offers a gentle option, mainly due to a reduction in the impact force during the isolation process, minimizing the effect on cell viability. On the other hand, the fact that NEs are internalized and degraded by cells over time facilitates the cell detachment process when compared with the standardized harvesting cell process from culture plates.

## 5. Conclusions

We successfully formulated Pept-NEs for breast cancer CTC targeting, through the use of a bottom-up strategy which provides the advantages of controlling the composition of NEs and the surface peptide concentration.

We studied the immobilization of Pept-NEs on chip surfaces and their specific binding to cell membrane proteins by QCM-D. A cytotoxicity analysis showed that the immobilized Pept-NEs did not affect cell viability. Therefore, the use of Pept-NEs appears to be an interesting option for the development of a potential isolation technology, due to the possibility that on-chip culturing reduces the risk of cell loss. This is related to the fact that our technique avoids the need for multi-step procedures associated with off-chip re-culturing. However, further work must be carried out to optimize the use of Pept-NEs-functionalized chips for the isolation of CTCs.

## Figures and Tables

**Figure 1 bioengineering-09-00380-f001:**
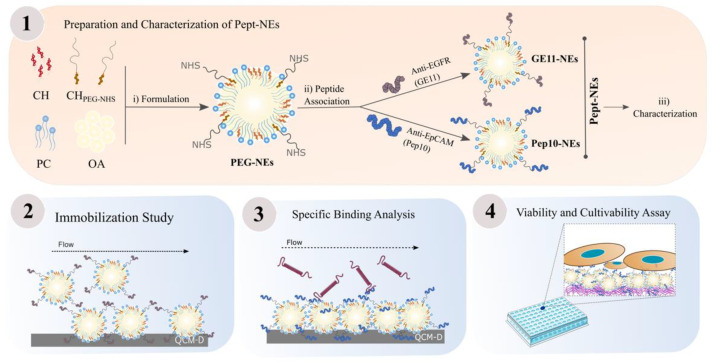
Workflow for the Development and Characterization of Peptide-functionalized Nanoemulsions (Pept-NEs). (1) Preparation and characterization of Pept-NEs: (i) formulation of PEG-NEs, (ii) association with two peptides (GE11 and Pep10) to obtain GE11-NEs and Pep10-NEs, and (iii) characterization. (2) Study of the immobilization of NEs on surfaces by Quartz Crystal Microbalance with Dissipation Monitoring (QCM-D). (3) Study of the specific binding ability of Pept-NEs to their targeted protein by QCM-D. (4) Cell viability and cultivability study for 9 days on Pept-NEs functionalized plates.

**Figure 2 bioengineering-09-00380-f002:**
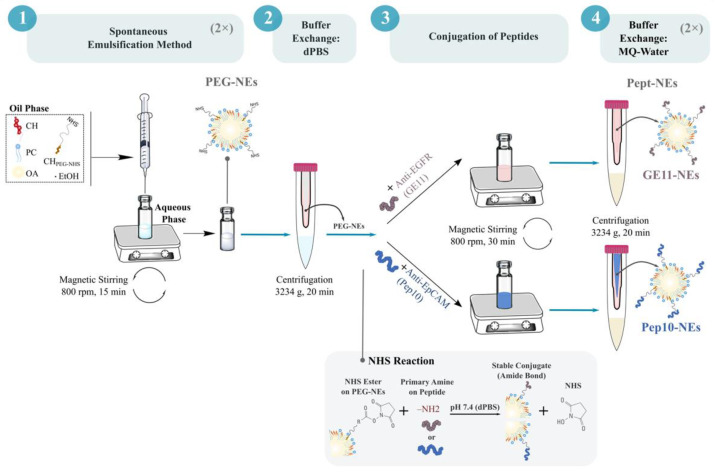
Schematic representation of the formulation procedure to obtain the Pept-NEs (GE11-NEs and Pep10-NEs). The four main steps in the procedure are: (1) spontaneous emulsification, performed twice (2×) to obtain 2 mL of PEG-NEs; (2) buffer exchange from MQ-water to dPBS; (3) conjugation of peptides using 1 mL of PEG-NEs for each peptide (the chemical reaction is represented in the grey box), and (4) performing a second buffer exchange, from dPBS to MQ-water, followed by storage.

**Figure 3 bioengineering-09-00380-f003:**
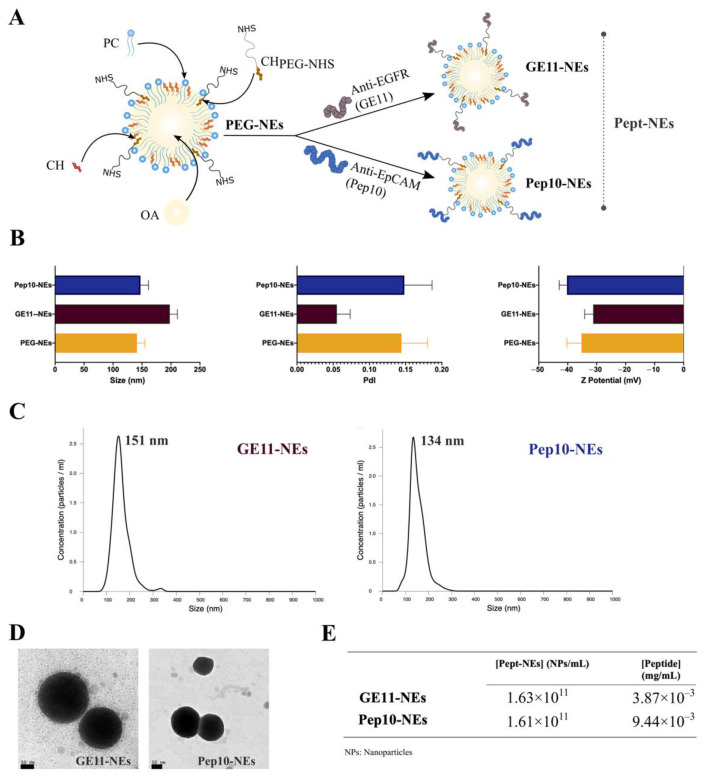
Characterization of PEG-NEs, GE11-NEs, and Pep10-NEs. (**A**) Scheme for the formulation of PEG-NEs made of phosphatidylcholine (PC), cholesterol (CH), oleic a(OA), and cholesterol-PEG with N-HydroxySuccinimide ester-PEG modification (CH_PEG-NHS_) and the association of peptides to obtain Pept-NEs (GE11-NEs and Pep10-NEs). (**B**) Hydrodynamic size (nm), polydispersity index (PdI), and ζ-Potential (mV) measurements. Values represent mean ± SD (*n* = 10). (**C**) Sizes of Pept-NEs as determined by Nanoparticle Tracking Analysis (NTA). (**D**) TEM images of GE11-NEs and Pep10-Nes. (**E**) I Pept-NEs concentrations (NPs/mL) and concentrations of the peptides in 1 mL formulation (mg/mL).

**Figure 4 bioengineering-09-00380-f004:**
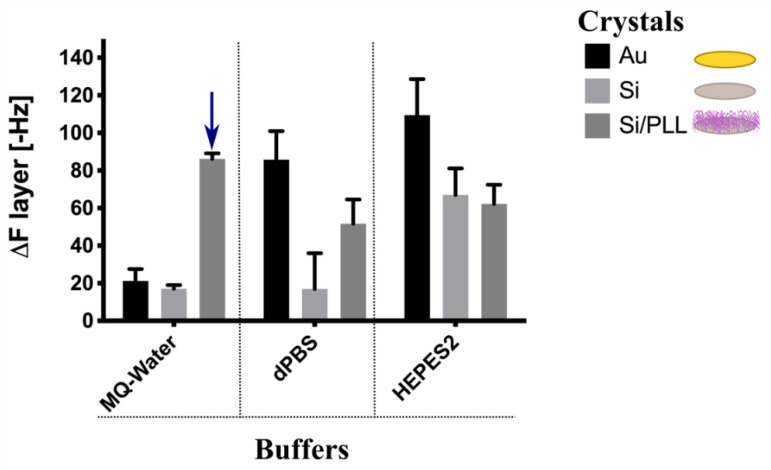
Immobilization analysis on surfaces of PEG-NEs. QCM-D frequency changes upon binding of PEG-NEs to different surfaces (crystals used: gold (Au), silica (Si), and silica/polylysine; Si/PLL) using different buffers (MQ-water, dPBS, and HEPES2). Values represent mean ± SD (*n* = 2). Blue arrow indicates the condition chosen.

**Figure 5 bioengineering-09-00380-f005:**
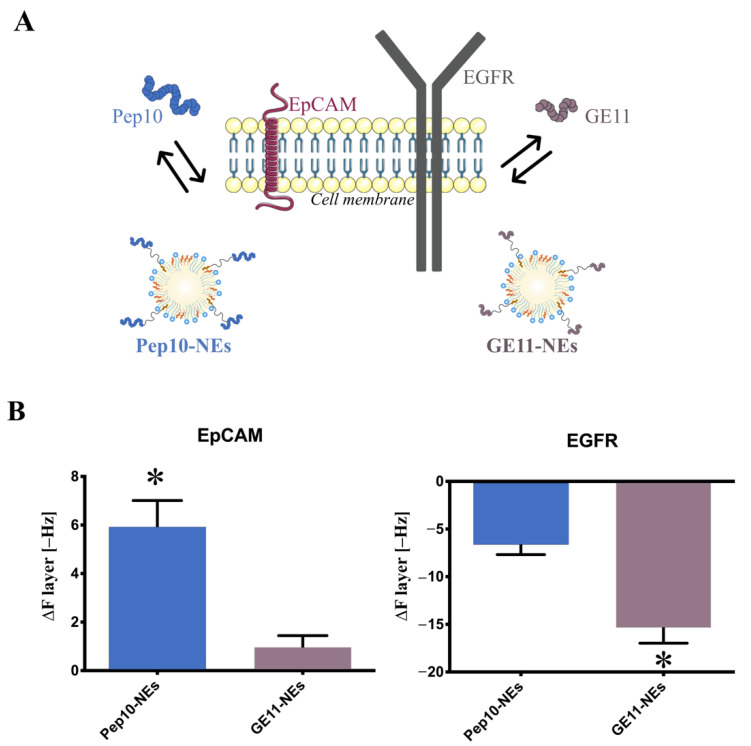
Specific binding analysis of Pept-NEs at the protein level. (**A**) Diagram of targeting abilities of Pept-NEs (Pep10-NEs and GE11-NEs), ligands (GE11 and Pep10 peptides) and their receptors (EGFR and EpCAM), respectively. (**B**) QCM-D frequency changes upon binding of proteins (EpCAM and EGFR) to Pept-NEs (Pep10-NEs and GE11-NEs). Values represent mean ± SD (*n* = 2). Statical differences are represented as * (*p* < 0.05).

**Figure 6 bioengineering-09-00380-f006:**
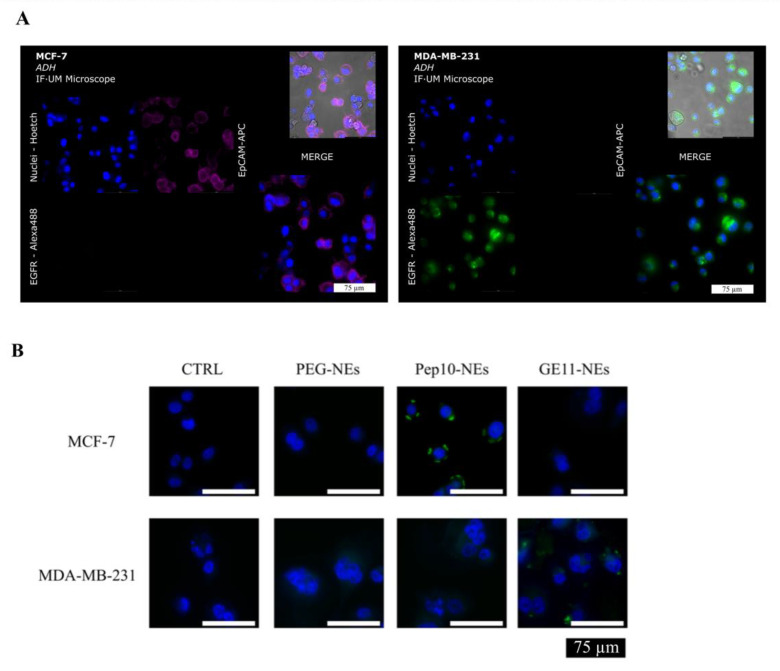
Specific binding analysis of Pept-NEs. (**A**) Specific expression of EpCAM by MCF-7 cells and EGFR by MDA-MB-231 cells. (**B**) Binding analysis by fluorescence microscopy of PEG-NEs and Pept-NEs to MDA-MB-231 and MCF-7. Negative controls (CTRL) show cells without PEG-NEs or Pept-NEs incubation. Scale bar represents 75 µm. Fluorescent images, blue arising from DAPI-Stained nuclei (cell nuclei) and green from fluorescent phosphatidylcholine introduced during the formulation of NEs.

**Figure 7 bioengineering-09-00380-f007:**
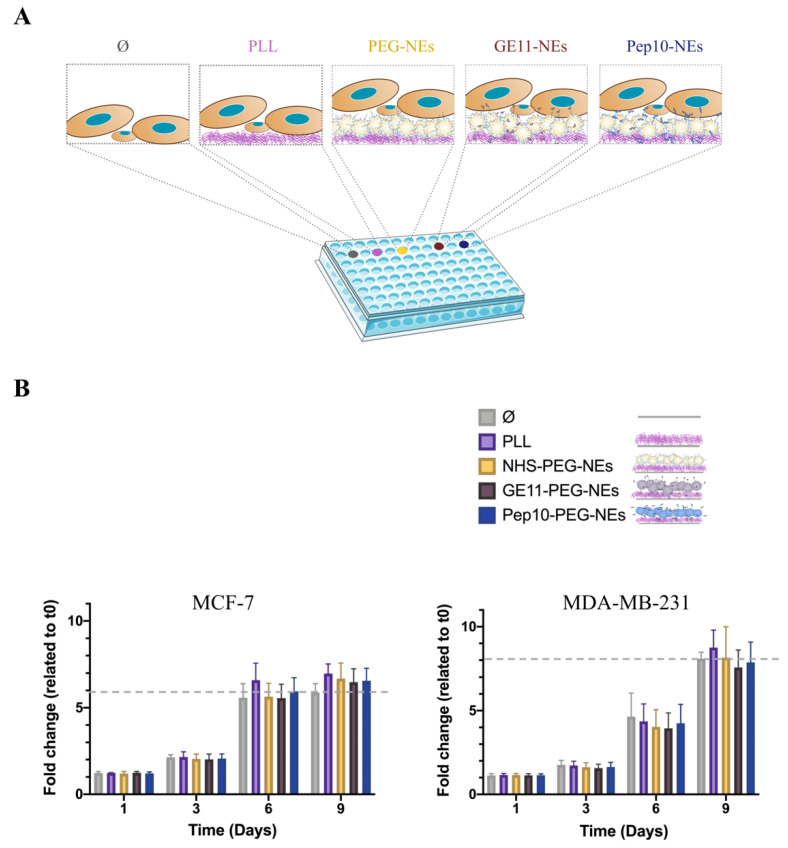
Cell viability Assay. (**A**) Scheme of the experimental setup. Breast cancer cells were in contact with different surfaces during 9 days of culture in a 96-well plate. Ø denotes the cell cultures in wells without pre-treatment; the rest were pre-treated with PLL. In the case of PEG-NEs and Pept-NEs conditions, formulations were added after PLL pre-treatment. (**B**) Viability results of MCF-7 and MDA-MB-231 cells incubated with PLL, PEG-NEs and Pept-NEs (GE11-NEs and Pept10-NEs) immobilized on plates. The results are expressed in fold-change values normalized to the corresponding control (Ø). Values represent mean ± SD (*n* = 3).

## Data Availability

The data presented in this study are available in this article and Appendix A.

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
