# Peer review of "Peptide-Functionalized Nanoemulsions as a Promising Tool for Isolation and Ex Vivo Culture of Circulating Tumor Cells"

_bioengineering, 2022, doi:10.3390/bioengineering9080380_

Round 1

Reviewer 1 Report

please find attached

Author Response

Response to Reviewer 1 Comments

Experimental

  1. Expression of surface markers (EpCAM, EGFR) should be verified on protein level. Since expression levels can change in cell lines, it is not sufficient to just refer to a publication from 2016! By the way, citation is wrong here (227?) or completely missing.

We apologize to the reviewer for the mistake; we have corrected it in the new version. We believed that the expression of EpCAM or EGFR in commercial cell lines is well described in literature, we have added new references 36 and 37 in the manuscript (DOI: 10.2174/1566524021666210729144713, DOI: 10.3390/ijms23116122). Besides, we have been used this cell line as a positive control in our laboratory for several years, also we have added a new figure to the final version (Figure 6, page 12) showing the different expression profiles by immunofluorescence.

  1. Design of experiments to test for specific binding of cells to Pept-NE is poor, and quite artificial: used cell concentrations are too high to reflect the biological situation of CTCs in blood, and cells were fixed before interaction.

We agree with the reviewer that the used numbers did not mimic the real situation of CTC in blood; however, the objective of this experiment was to prove the capability of the pept-NEs to interact specifically with their target cells. Performing a more realistic experiment with lower cell numbers (mimicking CTCs situation) and co-culturing them is of interest and we would like to do it in the future. Nevertheless, the scope of this manuscript is to prove that our system works for targeting in a specific manner. This aspect has now be clarified in the manuscript (page 16, line marked in yellow).

  1. Same for capture efficiency- 300,000 cells/ml are not a realistic concentration for CTCs in blood! Experiments should be performed with less cells, also including different “backgrounds”, such as comparison of PBS, FBS, media, and blood. Especially the unspecific capture of white blood cells is a major challenge in this field, and should be tested to prove applicability of the setting in a realistic setting.

We agree with the referee, the number of cells seeded is not realistic when compared with the number of CTCs that we will find in blood. Nevertheless, as we explained before, the objective of this experiment is to test if pept-NEs have the capability of targeting interaction. Additionally, related to the experimental design there are a series of limitations concerning the minimum number of cells required to be later collected and detected by a flow cytometer. We have added a paragraph to clarify this issue in the manuscript (page 16, lane marked in yellow). We understand what the referee says about to use different backgrounds, however this experiment was performed in cell media since is the preliminary work that will be completed with more experiments in the future.

  1. Capture efficiency is shown in cell concentration (% of PLL). It would be also important to show how many cells (%) were captured compared to initial amount of incubated cells. Capture efficiencies of over 90% have been reported for antibody-bead systems, eg [1]. If their system does not achieve comparable results, the authors at least have to mention and discuss that fact.

Thank you for the comment; we have added this issue in the discussion section in the new version of the manuscript (page 16, lanes marked in green).

  1. Cell viability assay: authors claim to mimic the situation of CTCs in a blood sample from patients” which is not true! Cultivation should be tested with lower cell numbers.

We agree with the referee about that we do not really mimic the CTCs situation, however as the manuscript mentions, “To mimic the situation of CTCs in a blood sample from patients, the smallest number of cells that the CCK-8 can detect were seeded (1,000 cells/well)”. Therefore, we needed to perform the experiment with these amount of cells to be able to measure cell viability. We choose this assay since in our experience we realized that CCK-8 is more reliable for cell viability than alamarBlue where we observe a lot of variability in these kind of experiments. We also perform this approximation in our previous work “Nanoemulsions to support ex vivo cell culture of breast cancer circulating tumor cells” materials today chemistry, 16, June 2020, 100265. In these publication we set up the conditions of cultivability and NEs concentration using cells lines and these amount of cells and then we extrapolate then to real CTCs as we publish in the work “Short-term ex vivo culture of CTCs from advance breast cancer patients: clinical implications”. Cancers13(11), 2668. The final application for these Pept-NEs coating chip surfaces will be the isolation and culture of CTCs, however the objective of this work was to prove the capability of Pept-NEs to interact with their targets, since the capability of our NEs to improve cultivability of CTCs was already proved in our previous works published in materials today chemistry and Cancers.

  1. Formal

Structure of the manuscript: according to the template, manuscript should be structured into 3. Results, 4. Discussion, and 5. Conclusion (now discussion appears twice? And authors did not really “discuss” the results in section 3). On the other hand, the “excessive” sub-structured results section (eg 3.3.1 and 3.3.2 should be combined in one paragraph, including Figures) is not required, and should be revised. This would significantly improve clarity for the reader.

We thank to the reviewer for the comment and we have corrected both the structure and the sub-structure. We also have restructured the manuscript adding a Discussion and Conclusion.

Numbering of figures is confusing, after figure 6 comes figure 9, then 7, figure 8 does not exist at all!

We really apologize for this mistake, we have solved it.

Several typos and non-sense sentences, eg. p. 15, l 528f. English/writing needs improvement!

We have revised the manuscript and improved it

Abstract should be revised summarizing more clearly the overall results of the study, as well as giving a conclusion. Now it reads more like a summary of performed experiments. “NE” is not explained at first mention.

We appreciate the comment and we have improve the abstract

Resolution of figures is low and should be improved.

We have improved the resolution of the figures

Reference

  1. Gribko, A., et al., IsoMAG—An Automated System for the Immunomagnetic Isolation of Squamous Cell Carcinoma-Derived Circulating Tumor Cells. Diagnostics, 2021. 11(11): p. 2040.

Reviewer 2 Report

In this work, the author designed Pep-NEs with affinity to breast cancer CTC and demonstrated Pep-Nes can be immobilized on chips for CTC isolation. The concept of this work is very good, and it could provide a modular way for on-chip cell isolation as different type of peptides can be integrated into the system developed here. However, some presentations of the results are confusing, and they need to be clarified before publication.

1.  In line 100, the author claimed “Additionally, he proved how Pep10 presented better binding affinity than EpCAM anti-body for EpCAM recognition (Pep10 KD is 1.98 x 10-9 mol/L and EpCAM antibody KD is 2.69 x 10-10 mol/L)”. Apparently, EpCAM antibody has a smaller Kd and indicates that EpCAM presents better binding affinity than Pep10.

2. Figure 4 is very confusing to read and hard to capture the adsorption and desorption value.

3. From Figure 5B, a desorption phenomenon is seen in for the binding of EGFR. Therefore, the statement of “In any case, GE11-NEs presents the higher desorption effect, suggesting a strong interaction with its targeted protein (EGFR)” needs more supports. A control experience using bare PLL-coated silica crystal or PEG-Nes coated chips as controls might provide more insights here.

4. Programmer errors need more attention. For example, line 418, “whas” typo.

Author Response

Response to Reviewer 2 Comments

In this work, the author designed Pep-NEs with affinity to breast cancer CTC and demonstrated Pep-Nes can be immobilized on chips for CTC isolation. The concept of this work is very good, and it could provide a modular way for on-chip cell isolation as different type of peptides can be integrated into the system developed here. However, some presentations of the results are confusing, and they need to be clarified before publication.

  1.  In line 100, the author claimed “Additionally, he proved how Pep10 presented better binding affinity than EpCAM anti-body for EpCAM recognition (Pep10 KD is 1.98 x 10-9 mol/L and EpCAM antibody KD is 2.69 x 10-10 mol/L)”. Apparently, EpCAM antibody has a smaller Kd and indicates that EpCAM presents better binding affinity than Pep10.

We would like to thank to the reviewer for noticed this mistake. The paper statement says: “KD were calculated from the measured ka and kd (KD = kd/ka), where a lower value of KD corresponds to a stronger binding affinity. As listed in Table 1, Pep10 shows comparable binding affinity (1.98 × 10-9 mol · L-1) to that of anti-EpCAM (2.69×10-10mol · L-1), which is in agreement with the above FCM data. This result indicates that Pep10 can be the possible candidate in CTC isolation as a complementary surface modification agent.

Accordingly, we have modified this statement in the manuscript.

  1. Figure 4 is very confusing to read and hard to capture the adsorption and desorption value.

We have change the resolution of the figure

  1. From Figure 5B, a desorption phenomenon is seen in for the binding of EGFR. Therefore, the statement of “In any case, GE11-NEs presents the higher desorption effect, suggesting a strong interaction with its targeted protein (EGFR)” needs more supports. A control experience using bare PLL-coated silica crystal or PEG-Nes coated chips ascontrols might provide more insights here.

We thank the comment of the reviewer, however the aim of this experiment was to compare the affinity of the ligands for their target proteins. We did not use PLL as a control since we understood that the binding between PLL and the proteins EGFR and EpCAM will be thought ionic interactions.  Therefore these data would not be comparable with the results that we expect through the interaction between pept10 and EpCAM and GE11 and EGFR.

Related with the "PEG-NEs coated chips as controls" suggestion, a similar experiment is represented in figure 7, where a chip was coated with Pept-NEs to analyse its interaction with the proteins expressed by cell. Additionally we have modified the manuscript and added a discussion section where we explained better and with new references the desorption process for EGFR (page 15, lanes marked in grey)

  1. Programmer errors need more attention. For example, line 418, “whas” typo.

We apologize for these mistakes we have reviewed the manuscript to solve them

Round 2

Reviewer 1 Report

Unfortunately, the authors did not significantly change the previous manuscript! Comments were NOT addressed properly: it is not sufficient to just argue when additional experiments are needed! Neither the abstract (give a conclusion. “NE” is not explained at first mention), nor the results section was revised as suggested (just single words/sentences changed).

Concerning binding specificity and capture efficiency, as I stated before I still believe that with some additional data the whole study would gain more relevance! Of course, future studies are needed, but this preliminary study lacks important controls which are needed to justify future evaluation of this method!

Additionally, I still do not understand presentation of capture efficiency experiments! If it is % of PLL control then it means that for MDA-MB-231 cells capture is not specific at all?! Here, 100% and below means that the same amount of cells or lower were captured compared to uncoated control? Authors have to clarify that, the added paragraph in the discussion is not helpful (and also needs major formal revision! Words are not used correctly, eg interestedly, conscientious…). As already suggested, authors should also present amount of captured cells as % of initial cells. It is a big difference if 90%, or just 10% of initial cells were captured, whether specific or not! To give you an example: for MCF-7 about 200% of cells could be captured with GE11-NEs compared to control, that means 2-fold which sounds good. BUT if in the control just 5% of initial incubated cells were (unspecifically) captured, this means specific capture efficiency is “just” 10% for coated chip?! By the way that´s just re-analysis of existing data, and should be easily done!

There is still need for professional proof-reading service! In summary, the manuscript is not acceptable in its present form.
